# Strain Gauge Calibration for High Speed Weight-in-Motion Station

**DOI:** 10.3390/s24154845

**Published:** 2024-07-25

**Authors:** Agnieszka Socha, Jacek Izydorczyk

**Affiliations:** 1APM PRO sp. z o.o., 43-300 Bielsko-Biała, Poland; agnieszka.socha@apm.pl; 2Faculty of Automation, Electronics and Computer Science, Silesian University of Technology, 44-100 Gliwice, Poland

**Keywords:** high speed weigh-in-motion (HSWIM), calibration of weigh-in-motion stations, calibration function, strain gauges, direct enforcement

## Abstract

The development of systems for weighing vehicles in motion aims to introduce systems allowing automatic enforcement of regulations. HSWIM (high speed weight-in-motion) systems enable measurement of a mass of vehicles passing through a measurement station without disturbing the traffic flow. This article focuses on the calibration of a weighing station for moving vehicles, where strain gauge sensors are used to measure pressures. A solution was proposed to replace the calibration coefficients with calibration functions. The analysis was performed for two methods of determining wheel loads: based on the maximum of the signal from strain gauge sensors and on a method using the field under the signal and the vehicle’s speed. Calibration functions were determined jointly for all test vehicles and separately for each of them. The use of a calibration function for a specific vehicle type made it possible to determine wheel pressure and gross weight with a level of accuracy that allowed the weigh-in-motion station to be classified as a direct enforcement system. The achieved improvement in the accuracy of weighing in motion did not require any interference with the measurement station. The proposed change in the method of calibration and, ultimately, determination of wheel loads required only a change in the algorithm for determining wheel loads.

## 1. Introduction

The United States established the first weigh-in-motion stations for vehicles in the 1950s. Initially, these systems were used primarily to collect data on the axle loads of vehicles to design road surfaces. Over time, as weigh-in-motion (WIM) systems evolved, they started being used to assess the fatigue of bridge structures and supervise heavy traffic [1].

The movement of vehicles with excessive loads causes damage to the road surface by creating ruts and defects. Nowadays in Poland overloaded vehicles account for 14% to 23% of all vehicles but are responsible for 35% to 70% of road surface deterioration. Among these vehicles, those with a total allowed weight of 40t contribute the most to road degradation. For the typing of overloaded vehicles that should be subjected to a stationary inspection by the Road Transport Inspectorate, pre-selective weigh-in-motion stations are used [2].

The precision of weigh-in-motion systems depends on the weighing error compared to standard values established through static measurements conducted during the station’s regular calibration. The advancement of WIM systems is progressing towards automated solutions, contingent on regulatory changes and improved system accuracy [3].

The ongoing research on weigh-in-motion systems aims to improve the precision of these stations to the level necessary to automatically impose penalties on overloaded vehicles and mitigate the impact of environmental factors on measurement results.

In the testing facility created by [4], they evaluated the precision of the systems using quartz sensors and bending plates. These sensors were positioned in four rows, enabling the pressure assessment of every axle to be conducted four times. The improvements in weighing precision were made by calculating the mean pressure values obtained from each row of sensors.

The authors of [5] suggest employing two separate sensor systems. One system is used to calculate the overall weight of the vehicles, while the second system’s measurement validates the initial measurement. This approach enables the precise identification of overloaded vehicles with exceptional accuracy (>99.8%), ensuring that vehicles of the appropriate weight are not misclassified as overloaded.

Algorithms are currently being developed to automatically assess the dependability of measured data. In [6], the authors suggest fuzzy logic as a method to determine the reliability of the measurements. To evaluate the system’s precision, confidence measures were applied to four factors influencing measurement accuracy: vehicle acceleration/deceleration, sensor location, vehicle speed adjusted for wind speed, and road surface temperature. Another approach, as detailed in [7], involves the creation of precision maps to define acceptable ranges of variation for factors that affect the precision of the measurement (such as speed, temperature, surface condition, and wind). This research focused on systems using quartz and piezoelectric sensors, with precision maps custom-made independently for each station.

Proposed solutions for automatically calibrating systems [8] are also suggested. This approach includes the ongoing calculation of the calibration factor using the initial axle loads of the specified vehicles as a reference. In addition, the author suggests incorporating temperature adjustments in the calibration factor determination process.

### 1.1. Classification of Weigh-in-Motion Systems

As stated in [9], the systems for weighing moving vehicles are categorised into:Statistical weights are used for gathering data on truck traffic. The permissible errors in measuring the total mass are 20% for class D + (20) and 25% for class D (25).Pre-selection weights are used to pre-select overloaded vehicles and to plan road and bridge maintenance. The allowable inaccuracies in determining the total mass are 10% (Class B(10)) and 15% (Class C(15)).Weights used for legal purposes include automatic ticketing and pre-selection of overloaded vehicles. The acceptable error margins in the calculation of the total weight are 5% (Class A (5)) and 7% (Class B + (7)).

Furthermore, systems for weighing moving vehicles can be categorized according to installation location. The most commonly used categories include the following:HSWIM (high-speed weigh-in-motion) systems are typically installed on public roadways where vehicles move at speeds allowed for a specific road category. These systems can record data for all vehicles passing through the measurement station. Various types of sensors, such as strain gauges, quartz, or piezoelectric, are employed to measure the pressure the passing vehicle applies. The accuracy of these installations ranges from ±5% to ±10% [1]. HSWIM systems are designed to weigh vehicles travelling from 30km/h to 130km/h and are commonly utilized for pre-selection and statistical purposes [9].LSWIM (low-speed WIM) systems are typically set up on specialized control platforms around 30–40 meters long. The precision of LSWIM systems typically falls within the range of ±3% to ±5% [10]. These systems employ plate sensors about 50cm wide and operate at speeds below 10km/h for weighing purposes. Consequently, the measurements obtained at LSWIM stations closely resemble those from static weighing [11].B-WIM (bridge WIM) systems are typically placed on bridges and viaducts to assess the strain induced on the structure by passing vehicles. These setups involve sensors distributed across the entire bridge or specific sections. The precision in determining the overall vehicle weight falls between ±5% to ±10%. By utilizing B-WIM systems, it becomes feasible to observe how passing vehicles affect the integrity of the bridge [1,12].

### 1.2. WIM Accuracy

The precision of weight-in-motion measurement systems can be influenced by various factors, which can be categorized into those associated with the vehicle itself, such as its speed, manner of passing through the measurement station, or the condition of the vehicle’s suspension. Environmental factors such as surface quality and temperature belong to another group. In addition, factors related to the system in place, such as the type and setup of sensors used and the algorithm for calculating the overall weight of the vehicle, are also significant. Therefore, the precision of the HWIM system is impacted by:Number of sensors. In multisensor setups, multiple weight measurements are taken for each wheel of the vehicle, allowing for a greater sample size for analysis. The weights of the individual axles and the total mass are calculated as the mean of the collected samples [3,13].Surface condition. Vehicle weighing-in-motion systems are usually set up on straight road segments where vehicles travel at a relatively constant speed. The positioning of pressure sensors on the road surface means that the measurement system comprises both the sensors and the road surface itself, making the surface damage directly affect the sensors’ functionality and the overall system operation. Road conditions must adhere to the specifications outlined in the COST-323 standard within a radius 50m before and a radius 25m after the measurement station. Lower surface quality is directly correlated with a decrease in the precision of the systems that can be deployed in that area [9].Ride dynamics. In [14], the authors investigated how a vehicle passing through a WIM station using strain gauge sensors can affect the results. They observed that driving on the right side of the road could lead to greater inaccuracies in determining the total weight of the vehicle. When vehicles hovered over sensors on the left side, the inaccuracies in total weight determination were similar to those in central hovering. If there was a change in the path taken through the measurement station, such as the switch from left to right, the total mass determination error did not exceed a certain threshold. However, there could be inaccuracies in determining the loads on individual axles.The velocity of a vehicle affects the functionality of its suspension system, affecting the pressure that the wheels exert on the road surface. Research in [15] showed that as the speed of the vehicle increases, there is an increase in the margin of error when determining the total weight. This situation could be mitigated by installing more sensors. In a separate study [14], the influence of speed variations on the data collected by strain gauge sensors was examined. The pressure of the front axle was found to be more significant during braking than driving at a constant speed. Changes in speed, such as acceleration or deceleration, result in modifications to the waveform of the recorded data, with the amplitude of the signal received from the strain gauge sensors increasing during acceleration. Two rows of strain gauge sensors were used to address inaccuracies that stem from speed fluctuations.Pavement temperature. The temperature range of the pavement containing the pressure sensors fluctuates between −30 °C and 60 °C [16]. With a rise in pavement temperature, the pavement’s stiffness diminishes, leading to an elevation in the pressure detected by the sensor. Temperature compensation has been suggested for systems employing piezoelectric sensors [15]. The discrepancy in calculating the total mass alters exponentially as the temperature rises.

## 2. The Research Testbed

The WIM testing facility used in our study can be found on DK44 in Mikołów, Poland. During the station design phase, surface examinations were performed, classifying it as class I according to the guidelines in [9]. The station is constructed to meet class B+(7) accuracy. The arrangement of the elements included in the measuring station is shown in Figure 1.

### 2.1. Construction of a Measuring Station

Strain gauge sensors—sensors 1.5m long and 7cm wide beams are utilized to measure the wheel and axle loads of vehicles passing through the WIM station. The structure of the sensor makes it adapted to measure only the vertical force acting on the sensor. Strain gauge sensors are not sensitive to lateral forces occurring through the action of the road surface on the sensor. Furthermore, the sensors used are equipped with an internal temperature compensation circuit so that they are not sensitive to changes in pavement temperature and the measurements are stable over time [17]. In addition, they are characterised by a low linearity error of <0.1± % FSO. These sensors are embedded on the road surface within grooves. The correctly leveled sensor is flooded with a dedicated road grout. To maintain optimal measurement conditions, the sensors are flush with the road surface, ensuring a level surface. Once the road grout has been installed and hardened, the sensors and the pavement are ground to remove any post-installation debris and to precisely align the levels. They come with a single power and signal cable which is connected to a data logger located in the control cabinet. The way the sensors are mounted means that the wheel of a passing vehicle has direct contact with the sensor. These sensors are known for their minimal linearity error and low sensitivity to temperature variations, which reduces their susceptibility to changes in surface temperature [18]. The traffic lane has four gauge strain gauge sensors in two rows four spaced 4m apart.

Piezoelectric polymer sensors—the piezoelectric sensors are employed to determine the point of contact between the tyre and the sensor, which indicates the path taken by the vehicle through the measurement station, as well as to identify instances of dual tyres [19]. The sensors take the form of metal rods installed on the road surface in specially cut grooves and then filled with resin. In contrast to strain gauge sensors, the wheel of the vehicle is not in direct contact with the sensor; the resin is the intermediate element in the pressure transfer. Consequently, they exhibit higher sensitivity to variations in surface temperature [13]. Compared to quartz sensors, piezoelectric polymer sensors have inferior metrological characteristics and are more susceptible to lateral forces [16]. Within the traffic lane, two sensors are placed at a $45^{\circ }$ angle to the traffic flow direction. The sensors are connected to a data logger in the control cabinet.

Induction loops—the loops are in charge of detecting the magnetic characteristics of a vehicle, triggering the generation of a data log, and activating ALPR (Automatic License Plate Recognition) and overview cameras. By analyzing the information collected by the loops, the speed of the vehicle and the time it passes through the measurement point [20] are calculated. The measurement system consists of a pair of 2.8m×1m induction loops.

ALPR camera—positioned on a pole around 20m from the approach loop, above the traffic lane. The camera has an image processing algorithm designed to scan the license plates of vehicles passing by [20].

Overview camera—positioned above the road, capturing overview images of vehicles in motion. Each vehicle image is assigned a unique timestamp. This identifier is also employed to merge the vehicle data log, utilizing information collected from all components of the framework [20].

The meteorological station and road surface condition sensor measure various atmospheric conditions, including air temperature, humidity, wind speed and direction, and the type and amount of precipitation. The surface condition sensor measures fundamental road surface characteristics like surface and subgrade temperatures, surface conditions, water film depth, and freezing temperatures [20].

The control cabinet houses various control and communication devices, including the iWIM data logger compatible with pressure sensors and induction loops.

### 2.2. Weighing-in-Motion

The block diagram of the weighing-in-motion system in Mikołów is shown in Figure 2. Strain gauge sensors, piezoelectric sensors, and inductive loops installed on the road surface are connected to the iWIM data logger.

The data collected by the components of the WIM station undergo processing in the analog and digital pathways of the iWIM data logger. In the analogue phase, the signals from the sensors are sampled at a frequency of several tens of kilohertz and undergo initial filtering. Subsequently, in the digital phase, an FPGA module processes the data, which comprises functional blocks such as a sensor input, decimation and compensation filters, a circular buffer, and an estimator. The sensor input module is responsible for receiving data from the measurement amplifiers. Signals received from sensors undergo filtration, the sampling frequency is reduced in the decimation filter, and any frequency discrepancies are corrected in the compensation filter. The data obtained are stored in a circular buffer and undergo preliminary analysis in the estimator module. Using data from induction loops and pressure sensors, the estimator predicts the vehicle speed and the entry and exit times at the measurement station. In addition, details such as the maximum signal values and vehicle ID are stored in the circular buffer. Specialized software identifies the signal peaks corresponding to the vehicle wheels passing and calculates the integral of the recorded signal. The exact vehicle speed and axle distances are determined by analyzing the location of the highest signal values. Subsequently, the iWIM data logger calculates the total weight of the vehicle on the basis of the applied algorithm. The ALPR camera, the overview camera, and the iWIM data logger transmit data to the WIM roadside unit on which the dedicated WIM system software is implemented. The software of the roadside unit includes modules that operate the specific devices, a time server, and a module of processing the collected data. The host software generates a data file that contains all recorded vehicle information. The precision of the collected data is then evaluated using the method outlined in [19,20]. A data record of a vehicle is encrypted and sent to another software module on the WIM server using the VPN. The data from the WIM server are then passed on to a graphical user interface (GUI) in an online application, where the collected data of each vehicle are visualised [19,20].

## 3. Measurements

The WIM stations undergo regular calibration following the standard set by the system manager [9,21]. The calibration of the test station is carried out periodically using reference vehicles with known weights [9]. As part of the tests carried out during calibration, static reference measurements are taken at the administrative weighing station and dynamic measurements are taken at the WIM station. The data collected are used for further analysis, which is described in the following sections.

### 3.1. Reference Measurements

Reference measurements were performed at an official weighing station using a certified scale by WITD staff (Provincial Road Transport Inspectorate—in Polish, Wojewódzki Inspektorat Transportu Drogowego). The scale used had a precision of ±25kg (for weights up to 2.5t) and ±50kg (for weights from 2.5t to 10t) according to OIML standards (International Organization of Legal Metrology), with a resolution of 50kg, allowing the weighing of wheels and axles. The official weighing station was equipped with WITD scales, elevated by 4.4cm, to ensure a level surface alignment between the scale and the measuring station plate during static weighing. Reference measurements were taken for the following vehicles of different gross vehicle weights (GVW):Two-axle truck with a GVW of 18t;Three-axle truck with a GVW of 26t;Five-axle truck with a semi-trailer with a GVW of 40t.

Each vehicle was equipped with a standard mass to ensure that the total weight nearly approached the maximum load capacity of the vehicles. The payload weight remained constant throughout the experiments. Figure 3 illustrates the standard mass used. The static measurements included recording the pressures exerted by the wheels of each vehicle and measuring the distances between the axles.

Each vehicle was weighed before and after test drives. Table 1 displays the mean results of the standard measurements $W_{S}$ for the left and right wheels and the total weight of the vehicle, which served as baseline values for subsequent analysis.

Furthermore, measurements were performed using a LP788 underlay scale with precision according to OIML standards: ±25kg (0–2500kg), ±50kg (2525–10,000 kg), and a resolution of 1kg. Two scales were calibrated on a hydraulic press at the District Office of Measures in Poznań, Poland. After the calibration process, the scales were subjected to a verification check. Standard loads were applied to each scale and their readings were recorded. The identified indication errors of the LP788 scales concerning the hydraulic press configuration, considering adjustments, ranged from −0.5% to 0.5%.

The LP788 scale is characterized by its low profile, standing at a height of 2.2cm. Upon placing the scales on the plates, there was an approximate level difference 2cm between the surface of the scale and the vehicle weighing station. Supports were used to raise the scales to address this level disparity. However, this set-up caused stability issues as the vehicle approached the scale. Consequently, some measurements were performed with the scale positioned directly in the trough. In this configuration, following the vehicle impact on the scales, the displayed value exhibited instability, with around 200kg per wheel fluctuations. Subsequently, the scales were repositioned before the trough to ensure that none of the wheels were within it during the measurement. Although this adjustment did not completely eliminate the level gap between the weighed axle and the other axles of the vehicle, traversing the scales helped to expedite the stabilization of the displayed value. The vehicles were weighed before and after test runs. The average results of the reference measurements $W_{S}$ for the left and right wheels, along with the total weights of the vehicles, are presented in Table 2.

Due to the variations in the experimental conditions, the measurements were compared using the precise Wilcoxon test [22]. All measurements performed at the weighing station were included in the analysis, comprising two measurements with a WITD scale and an LP788 scale. The test statistic value *V* for the measurements analyzed was 693, with a *p*-value for the null hypothesis of $3.1\times 10^{-05}$. The resulting low probability indicated significant differences between the two sets of measurements. In the case of the Wilcoxon rank sum test with continuity correction, the test statistic *W* was 916, and the corresponding *p*-value was 0.1332. Unlike in the previous case, although the test statistic value was high, the probability value was not low enough to suggest significant differences between the two sets of measurements. Based on the result of the Wilcoxon test without continuity correction and the methodology of measurement execution, specifically the scale settings, the measurements obtained with the WITD scale were selected as a reference for further analysis. Figure 4 compares the two static measurements, showing a median value of 3975kg for measurements with the WITD scale and 4118.5kg with the LP788 scale. Furthermore, total vehicle weights were higher for measurements with the LP788 scale, with all vehicles exceeding the GVW (Gross Vehicle Weight); see Table 1 and Table 2.

### 3.2. Dynamic Weighing

Trial operations were carried out in a moving vehicle weighing station to calibrate and test the system. Each vehicle traversed the weighing station 13 times. All trials were carried out on the same day. During each series of measurements, the test vehicles sequentially passed through the WIM station. The initial six runs were considered calibration runs and the subsequent ones were verification runs. Calibration runs were performed at an approximate speed of 50km/h, while verification runs were carried out at 40km/h, 50km/h, and 60km/h. The iWIM data logger documented all trips. Due to the conditions of the road traffic tests, two of the trial runs were incorrectly executed and deemed unsuitable for further analysis. Figure 5 establishes sensor labels:W1L—left sensor, first line;W2L—left sensor, second line;W1P—right sensor, first line;W2P—right sensor, second line.

The raw data from the iWIM data logger, captured for each sensor, were used for additional examination. Only the data documented by the strain gauges were used for subsequent analysis. Figure 6 shows an example of a strain gauge sensor signal waveform recorded for a triaxial vehicle. Each peak represents the signal obtained from a strain gauge sensor generated by the pressure of a wheel of a vehicle passing through the measurement station.

### 3.3. Establishing the Total Mass

Two techniques were used to determine the gross weight of the vehicles. The initial technique involved using the peak value of the signal, while the second technique utilized the area beneath the signal waveform and the speed of the vehicle [23]. In subsequent steps of the analysis, a modification of these techniques was implemented—instead of a single calibration coefficient *C*, calibration functions were established in the form of a linear or quadratic equation to determine the pressures of each wheel and subsequently the total weight of the vehicle. Two alternatives were suggested for the methods considered to determine the total weight of the vehicle:Establishing the overall mass via the signal’s maximum value.(a)A single calibration function for all examination vehicles.(b)Different calibration functions are assigned to each test vehicle.Estimation of the total vehicle mass by analyzing the area beneath the signal waveform.(a)A single calibration function applies to all vehicles.(b)Different calibration functions are designated for each vehicle for inspection.

#### 3.3.1. Signal Peak

The waveforms derived from each sensor, as shown in Figure 6, have a specific shape. The highest peak value of the signal was selected for analysis, representing the maximum value of the waveform. For each test vehicle, the waveforms were analyzed to determine the maximum value of each signal peak on all strain gauge sensors. Alternatively, the peak value of the signal can be calculated as the mean of, for example, 50 samples surrounding the maximum peak of the signal. The differences were noted to be minimal for the strain gauge beams; therefore, the peak values were used in the analysis.

The calibration functions for the entire set of test vehicles were determined using the reference values proposed by WITD and the maximum values of the signals from the recorded calibration runs. This is depicted in Figure 7, which presents static reference measurements for the peaks of the recorded signals.

In light of this, calibration functions were established for every sensor. In this instance, the calibration function was chosen to be a quadratic equation:(1)$$W_{D}=C_{1}\cdot {\left(\max\left(X_{i}\right)\right)}^{2}+C_{2}\cdot \max\left(X_{i}\right)+C_{3},$$
where:

$W_{D}$—the pressure applied on the sensor from the wheel of a traversing automobile;

*C*—coefficient, real number;

$\max(X_{i})$—peak signal value for each wheel as per the sensor under consideration.

Hence, the subsequent calibration functions were established for every sensor during the calibration trials:
(2a)$$\begin{array}{cc} W_{D1L} & =-0.00000985\cdot {\left(\max\left(X_{i}\right)\right)}^{2}+0.5200\cdot \max\left(X_{i}\right)-489.4\;\;\;\;\;\;\;\;\;\;\;\;\;\;\;\left(R^{2}=0.94\right), \end{array}$$
(2b)$$\begin{array}{cc} W_{D2L} & =-0.00001018\cdot {\left(\max\left(X_{i}\right)\right)}^{2}+0.5404\cdot \max\left(X_{i}\right)-935.7\;\;\;\;\;\;\;\;\;\;\;\;\;\;\;\left(R^{2}=0.94\right), \end{array}$$
(2c)$$\begin{array}{cc} W_{D1P} & =-0.00000837\cdot {\left(\max\left(X_{i}\right)\right)}^{2}+0.4711\cdot \max\left(X_{i}\right)-359.0\;\;\;\;\;\;\;\;\;\;\;\;\;\;\;\left(R^{2}=0.86\right), \end{array}$$
(2d)$$\begin{array}{cc} W_{D2P} & =-0.00000465\cdot {\left(\max\left(X_{i}\right)\right)}^{2}+0.3678\cdot \max\left(X_{i}\right)+77.28\;\;\;\;\;\;\;\;\;\;\;\;\;\;\;\left(R^{2}=0.87\right). \end{array}$$

Using calibration functions for each sensor, the pressures applied by the left and right wheels on two sensor lines were ascertained. Subsequently, the mean pressure from the left and right wheels was calculated to determine the total mass from the dynamic measurement and the discrepancy compared to the reference measurements. Both calibration and verification runs were computed. The results derived from these computations are displayed in the following section.

In the same way, calibration functions were established individually for each test vehicle. A linear approximation was used for the two-axle vehicle, whereas a polynomial approximation was used for the three-axle and five-axle vehicles. The diagrams that were used to determine the calibration functions are displayed in Figure 8.

Table 3 summarizes the assigned calibration functions. For each test vehicle, four of these functions were established. Using these, the pressures applied by the left and right wheels on both sensor lines were ascertained during each vehicle’s calibration and verification runs. Subsequently, the total masses of all test vehicles were calculated along with the errors compared to the reference values.

#### 3.3.2. The Area under the Signal Waveform

The signal received from the strain gauges was analyzed for every journey of each test vehicle, with the area below each peak calculated. The waveforms obtained facilitated the determination of the velocities of the left and right wheels on each axle of the vehicle. The speed was ascertained using the formula:(3)$$V=\frac{S}{t},$$
where:*S*—distance between the first and second line of sensors, 4m.*t*—travel time between the first and second sensor lines.

The duration of the travel of the wheel (vehicle) between two sensor lines, denoted by *t*, was calculated by dividing the count of the samples between the identified peaks of the signals on the initial and subsequent sensor lines by the sampling rate.
(4)$$t=\frac{l_{p}}{f},$$
where:$l_{p}$—quantity of instances between the peak of the signal from the initial and secondary sensor lines, for every axle of the vehicle.*f*—sampling frequency, 31.250kHz.

The calibration functions for the collective set of test vehicles were established using the reference values provided by WITD, together with the product of the calculated area under each peak (integral) and the velocity of each vehicle wheel (see Figure 9).

The calibration function for each sensor takes the shape of a linear equation:(5)$$W_{D}=\frac{C_{1}}{L}\cdot V\cdot \int_{t_{1}-t_{\Delta }}^{t_{2}+t_{\Delta }}\left(x\left(t\right)-b\left(t\right)\right)\;dx+C_{2},$$
where:$W_{D}$—the pressure applied to the sensor by the wheel of a vehicle as it passes over;$C_{1}$—coefficient, real number;*V*—velocity of the wheel traversing a specific sensor;*L*—sensor width;x(t)—signal generated by the load of a vehicle’s wheel as it passes by;b(t)—signal level at unloaded sensor;$t_{1},t_{2}$—signal threshold;$t_{1}-t_{\Delta }$—start of signal processing;$t_{2}+t_{\Delta }$—end of signal processing;$t_{\Delta }$—a constant proportional to the threshold level;$C_{2}$—constant.

In the case of calibration trials, each sensor was assigned the subsequent calibration equations:
(6a)$$\begin{array}{cc} W_{D1L} & =0.00001929\cdot V_{L}\cdot \int_{t_{1}-t_{\Delta }}^{t_{2}+t_{\Delta }}\left(x\left(t\right)-b\left(t\right)\right)\;dx+543.4\;\;\;\;\;\;\;\;\;\;\;\;\;\left(R^{2}=0.99\right), \end{array}$$
(6b)$$\begin{array}{cc} W_{D2L} & =0.00001772\cdot V_{L}\cdot \int_{t_{1}-t_{\Delta }}^{t_{2}+t_{\Delta }}\left(x\left(t\right)-b\left(t\right)\right)\;dx+456.8\;\;\;\;\;\;\;\;\;\;\;\;\;\left(R^{2}=0.99\right), \end{array}$$
(6c)$$\begin{array}{cc} W_{D1P} & =0.00002047\cdot V_{P}\cdot \int_{t_{1}-t_{\Delta }}^{t_{2}+t_{\Delta }}\left(x\left(t\right)-b\left(t\right)\right)\;dx+64.12\;\;\;\;\;\;\;\;\;\;\;\;\;\left(R^{2}=0.98\right), \end{array}$$
(6d)$$\begin{array}{cc} W_{D2P} & =0.00001864\cdot V_{P}\cdot \int_{t_{1}-t_{\Delta }}^{t_{2}+t_{\Delta }}\left(x\left(t\right)-b\left(t\right)\right)\;dx+120.0\;\;\;\;\;\;\;\;\;\;\;\;\;\left(R^{2}=0.98\right). \end{array}$$

Using calibration functions, we could determine the force of the left and right wheels on each axle across two sensor lines. Subsequently, we calculated the average pressures for each wheel. Based on these average forces from the left and right wheels, the overall weights of each test vehicle were established.

Calibration functions were established for each test vehicle, derived from the area beneath the signals and the speed of each vehicle. The OX axis of the graph displays the product of the area beneath the signal’s peak that reflects wheel pressure and speed. In contrast, the static WITD measurement is represented on the OY axis (see Figure 10). The points of each sensor on the graph were approximated using a linear equation, and the calibration functions obtained for all sensors are presented in Table 4.

The forces of each wheel on two sensor lines were calculated using collectively established calibration functions for all test vehicles. Subsequently, the mean load of each wheel and the total weight of the vehicle that traversed the measurement station were determined. The findings are discussed in the following section.

## 4. Results

### 4.1. Maximum Signal

Table 5 presents the results derived from using the standard calibration functions for each vehicle type, determined by the method that uses the highest signal value. In particular, these standard calibration functions recorded the most minor errors for a vehicle with five axes. The total weight of a vehicle with two axles tends to be undervalued relative to the benchmark value. In comparison, the total weight of a vehicle with three axles is typically over-valued. The discrepancies in the calculation of the total mass are pretty consistent across both the calibration and verification runs.

The weights of each wheel and axle of the test vehicles were calculated using calibration functions that were individually determined for each type of vehicle. The median of the total weight calculated using these calibration functions for each category of test vehicle, along with the error compared to the reference values, is shown in Table 6.

Inaccuracies in the calculation of the total weight using separate calibration functions for vehicles with two and three axles were notably lower than those using shared calibration functions (see Table 5). The discrepancy with respect to the benchmark measurement was decreased to 2%. Inaccuracies persisted at a magnitude comparable to that of a vehicle with five axles.

Figure 11 shows errors in determining pressures on each of the sensors for test vehicles using common and individual calibration functions. When calibration functions were used for a given vehicle type, errors in the pressure exerted on the sensor were reduced compared to standard calibration functions. In addition, errors in determining the pressure on the sensors for individual calibration functions were distributed more evenly relative to zero.

### 4.2. The Area under the Signal Waveform

The pressures and total masses for each sensor were measured, and the discrepancies compared to the reference values were calculated. The medians and discrepancies compared to the reference values were determined from the total masses of the calibration and verification runs (Table 7).

The total vehicle masses, computed using distinct calibration functions for each vehicle type, were subjected to a median determination for calibration and verification runs. The error in the reference measurement was also calculated for these values. Table 8 presents the results derived from these calculations.

In the case of determining the wheel loads and total mass based on the area under the signal waveform, errors relative to the reference measurement remained at a similar level for both standard and individual calibration functions shown in Table 7 and Table 8. Individual calibration functions allowed us to determine the total mass of the vehicle that passed through the measurement station with the highest precision.

Figure 12 illustrates the inaccuracies in determining the pressure on the sensor for each test vehicle, computed using shared and unique calibration functions. Adopting different calibration functions for each type of test vehicle resulted in fewer errors when determining wheel loads on the sensors compared to using shared functions. These errors are more uniformly dispersed around zero by applying individual calibration functions.

### 4.3. Vehicle Speed vs. Measurement Accuracy

The calibration of vehicle wheel pressure measurements must consider the vehicle’s speed and the intricate interactions between the vehicle’s supporting structure, suspension, wheel tire, road surface, and measuring beam. Creating a comprehensive model of the mechanical phenomena in this system is immensely intricate and demands extensive knowledge of vehicle and surface technical structures. Regrettably, there is a noticeable dearth of suitable studies in the literature, and the existing conclusions, e.g., [24,25,26], are not readily applicable to estimating measurement error. Hence, this study aims to experimentally determine the essential dependencies, despite the high cost and logistical complexity involved. The results obtained only permit estimating the maximum error within the tested speed range, as numerous uncontrollable variables influence the measurement. These variables encompass the vehicle’s axis distance from the transverse axis of the measuring beam, the angle between the vehicle’s movement path and the transverse axis of the beam, technical structure details of the vehicle, the vehicle’s technical condition, the road surface’s technical condition near the measuring beam, and weather conditions.

The test vehicles operated on a road with a maximum speed of 50km/h, adhering to normal traffic conditions. The evaluation was conducted at speeds of 40, 50, and 60km/h to determine the maximum errors in axle load determination. Please refer to Table 9 and Table 10 for detailed results.

The first method of determining pressures leaves the errors for different speeds at a similar level. However, using vehicle type-dependent calibration functions significantly reduced these errors for each test vehicle. 

### 4.4. Evaluating Different Techniques for Measuring the Pressure Exerted on the Sensor

The classification of the accuracy of the vehicle weighing system was established by considering the inaccuracies in calculating the total mass of the vehicle at the measurement station and the inaccuracies in calculating the axle load of the vehicle [9]. The pressure of each vehicle axle was determined as the sum of the average wheel pressure values of each axle measured on the two lines of sensors. Inaccuracies in the axle load calculation compared to the reference value for each method were used to determine the pressures applied to the sensors. These inaccuracies for each test vehicle are depicted in Figure 13.

For both methods of determining axle loads, when vehicle specific calibration functions were used, the axle load determination errors was smaller than for common calibration functions. For the method based on the peak value of the sensor signal, the errors on the axle remained at different levels depending on the calibration function adopted. For axle loads determined using the field under the signal waveform, the errors on the axle remained at similar levels for both types of calibration function.

As per [9], for systems of pre-selection and direct enforcement, the allowable inaccuracies in calculating pressures and overall mass are:Preselection:-Total weight: 7%;-Single axle load: 11%;-Load per axle of axle group: 14%;Direct enforcement:-Total weight: 5%;-Single axle load: 8%;-Load per axle of axle group: 10%.

For the method of determining the loads and the total mass using the maximum signal from the sensors, the errors obtained for the loads on the axle and the axle group are included in Table 11.

For each of the test vehicles, the error in determining the total mass relative to the reference value was calculated (Figure 14).

The errors in determining the total mass for test vehicles using common calibration functions varied within the following limits:Two-axle vehicle: from −11% to −8%;Three-axle vehicle: from 3% to 7%;Five-axle vehicle: from −1% to 3%.

However, when calibration functions were used for a given vehicle type, these errors varied within the following limits:Two-axle vehicle: from −2% to 3%;Three-axle vehicle: from −2% to 2%;Five-axle vehicle: from −1% to 2%.

The pressures determined using the maximum signal and the common calibration functions classified the system into statistical systems. Due to the use of calibration functions for a given vehicle type, errors in determining pressures and total weight were reduced to the level that would classify the station as an automatic system.

For the method using speed and the area under the signal waveform, the maximum error values for a single axis and a group of axles are summarized in Table 12.

The errors in determining the total mass of the test vehicles calculated using common calibration functions and for a given vehicle type are shown in Figure 15.

In the method using the field under the signal from the sensors, the error in determining the total mass of vehicles was at a similar level for common calibration functions and for a given vehicle type. For both methods of determining the wheel loads and total weight, the weigh-in-motion system was classified as an automatic system. However, Figure 15 shows that the errors in determining the total mass were more evenly distributed relative to zero for the calibration functions for a given vehicle type than for common calibration functions.

The results obtained for the method using the field under the signal waveform to determine wheel loads were compared with data obtained from the pre-selection WIM station in Mikołów, located just behind the test stand, in the same lane. The same system of strain gauge sensors was installed at the WIM preselection station, owned by GDDKiA (General Directorate for National Roads and Motorways—in Polish: Generalna Dyrekcja Dróg Krajowych i Autostrad), as at the test station (Figure 16).

At the preselection weighing station for moving vehicles in Mikołów, the highest errors in determining the total mass were obtained for a three-axle vehicle. The comparison of the results obtained using the calibration functions with the preselection system showed that replacing the calibration coefficients determined for each sensor with calibration functions kept the errors in determining the total mass for each type of test vehicle at a similar level. The accuracy of the system was improved at the level of the algorithm implemented to calculate the pressure exerted by a vehicle passing through the measurement station; there was no need to equip the HSWIM station with additional sensors.

## 5. Conclusions

Weigh-in-motion systems, and in particular HSWIM systems, are an element that allows for the identification of overloaded vehicles from the traffic flow. The development of on-the-go weighing systems towards automatic mandates requires the appropriate selection of an algorithm for assessing the reliability of measurements and increasing the measurement accuracy to the required level. An extremely important element is the appropriate determination of the reference level, because it is on its basis that calibration functions are determined. The accuracy of the static measurement is 2%. An additional factor affecting the accuracy of this measurement is the time that has elapsed since the portable wheel scales were calibrated. The degree of degradation of the administrative weighing station can also distort the measurement, through the difference in levels between the weighed axis and the other axes. During the tests, a difference in 2 cm levels caused significant differences between the static measurements.

Figure 11 and Figure 12 show the errors of determining the wheel pressures on the sensor. It can be seen in these figures that there was a scatter of errors, i.e., some of the errors were positive and some negative. Determining the axle pressures as averages of the measurements on the two lines resulted in the errors in the determination of the axle pressures (Figure 13) and the total vehicle weight (Figure 14 and Figure 15) being significantly smaller than the error of sensor pressures. Implementing calibration functions for the vehicle type stabilised the wheel pressure determination errors on the sensor. Increasing the measurement accuracy on the sensors will enable the detection of vehicles that are unevenly loaded.

The determination of wheel pressures using the peak value of the signal does not take speed into account in the calculation and is used for systems where a high degree of accuracy is not required. For more accurate systems, e.g., pre-selection, the second method is used, where the calculation takes vehicle speed into account.

The road surface in which the load sensors are installed is a part of the measuring system that is constantly degraded by vehicle traffic. An increase in the pressure error value of a sensor or a pair of sensors, e.g., the left sensors, can be caused by the fluctuations of a vehicle passing through ruts in the roadway. When comparing the maximum values of axle load errors at the tested speeds (see Table 9 and Table 10), it was evident that the differences for the tested speeds were approximately 1%. Furthermore, the second method of determining pressures, involving the utilization of the area under the signal waveform, considered the vehicle speed in the equations, leading to a more comprehensive analysis. It is crucial to acknowledge that under regular road traffic conditions, conducting measurements is subject to limitations imposed by other road users.

The layout of the pressure sensor at the tested station is used in pre-selection systems, while a method using the field under the sensor waveform and the vehicle speed is used to determine the pressures. The introduction of calibration functions instead of the coefficients proposed in the article made it possible to increase the accuracy of determining the pressures of the wheels and ultimately the total mass of the test vehicle. The test vehicles used had varying loads on individual axles. By employing calibration functions, we were able to precisely adjust the pressure equation to suit each vehicle type, resulting in accurate pressure determination and total weight measurement across all vehicles. Additionally, the number of wheels impacted the number of samples used to determine standard calibration functions. The most favourable outcomes were achieved for a five-axle vehicle. The utilization of type-specific calibration functions ensured consistent accuracy in determining axle pressures for all vehicles.

Due to the use of calibration functions for a given vehicle type, errors in determining the total mass remained at a similar level for each type of test vehicle. The improvement achieved in the accuracy of determining the axle loads and total weight allowed the tested station to be classified as an automatic system. An accuracy improvement was achieved for both the method using the maximum signal from the sensor and the method based on the field under the signal and the vehicle speed. The proposed solution does not require any interference with the measurement station.

The selection of the calibration function to determine the wheel load of a vehicle passing through the measurement station should be combined with the vehicle classification algorithm, which will cause the errors in determining the loads of individual axles and the total mass to be the smallest. The vehicle passing through the measurement station should be classified, and then the station software should determine the area under each of the peaks of the signal received from the strain gauge and the speed of the vehicles. After receiving information about the vehicle class, the mass determination algorithm should implement an appropriate calibration function for a specific vehicle type. The pressures determined in this way should be used to determine the loads of each axle and the total mass. The selection of the calibration function for the vehicle type should be verified by the measurement reliability assessment algorithm implemented in the master system in order to reject incorrectly classified vehicles and not expose drivers to unjustified inspections.

## Figures and Tables

**Figure 1 sensors-24-04845-f001:**
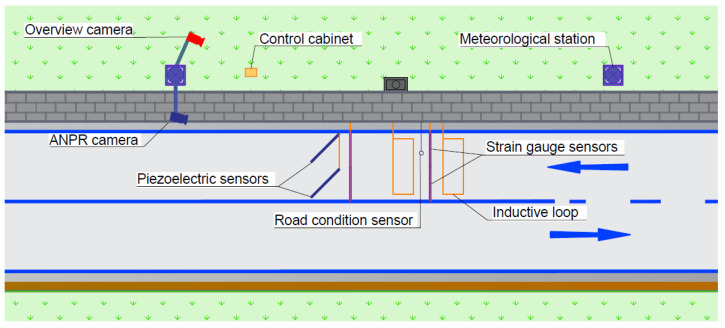
The research testbed—WIM station in Mikołów, Poland.

**Figure 2 sensors-24-04845-f002:**
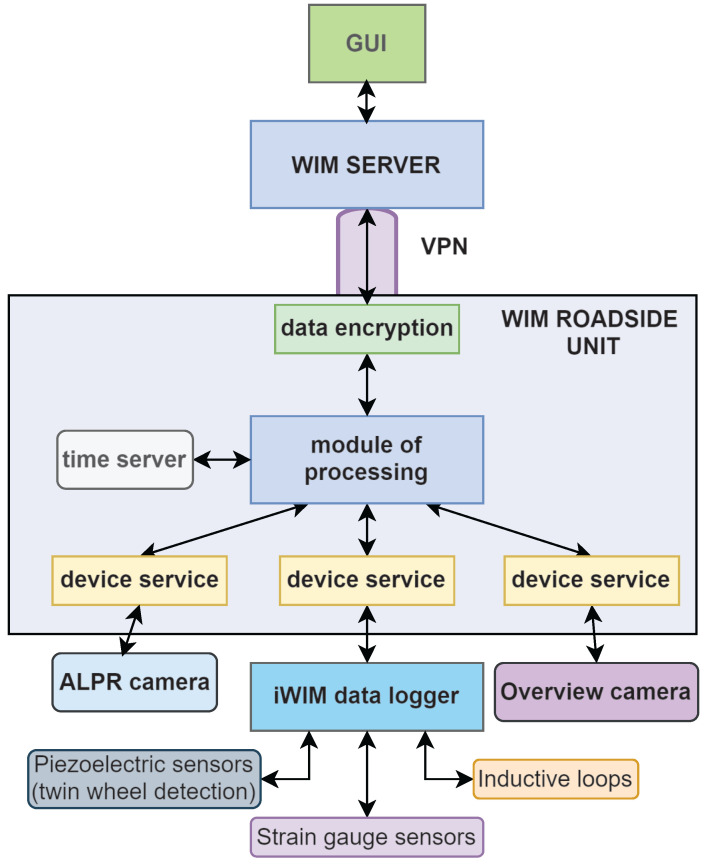
Block diagram of WIM system in Mikołów [20].

**Figure 3 sensors-24-04845-f003:**
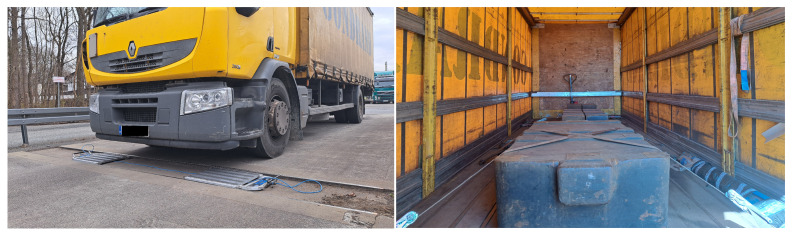
A vehicle with two axles used for reference measurements and a standard for mass.

**Figure 4 sensors-24-04845-f004:**
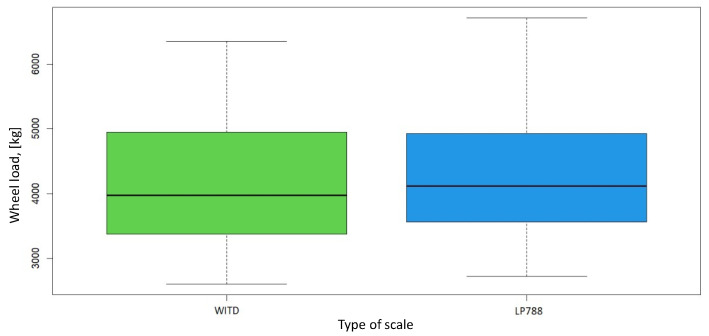
Comparison of static measurements at an administrative weighing station.

**Figure 5 sensors-24-04845-f005:**
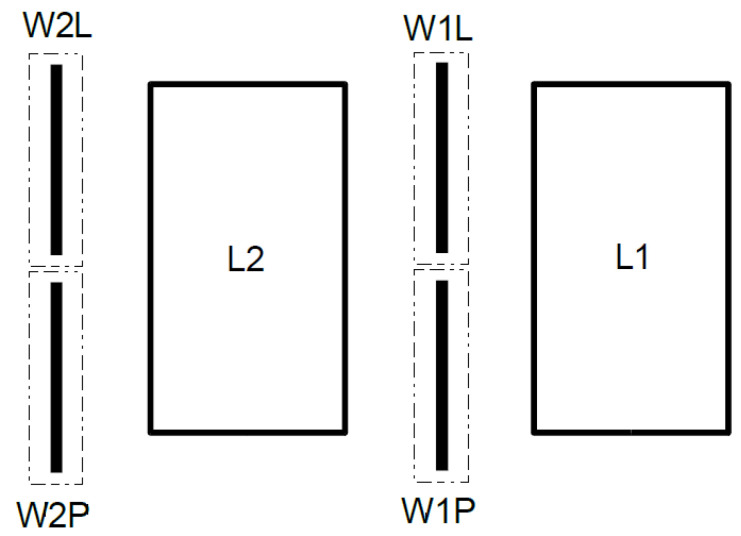
Layout of sensors connected to iWIM data logger.

**Figure 6 sensors-24-04845-f006:**
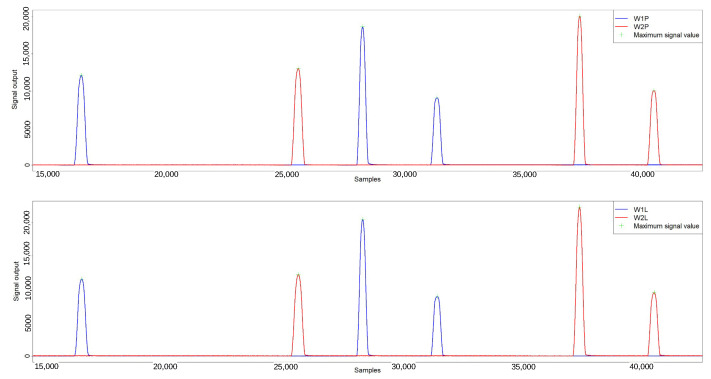
Signal waveform from two sensor lines for a three-axle vehicle.The top graph shows the signal from the right sensors, the bottom from the left sensors. Blue is the signal from the first line of sensors, red is the signal from the second line of sensors.

**Figure 7 sensors-24-04845-f007:**
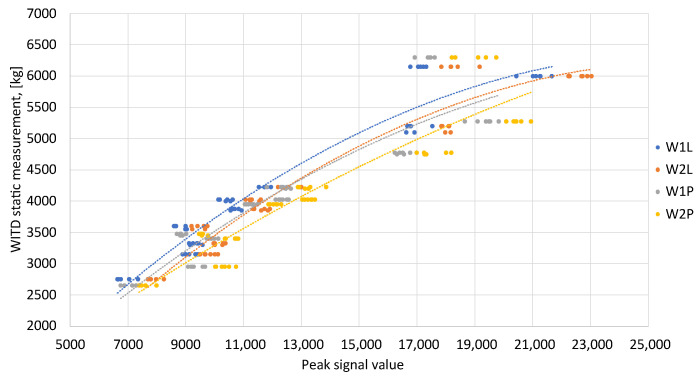
Establishing calibration functions for the complete set of examination vehicles. The lines represent interpolation by a polynomial.

**Figure 8 sensors-24-04845-f008:**
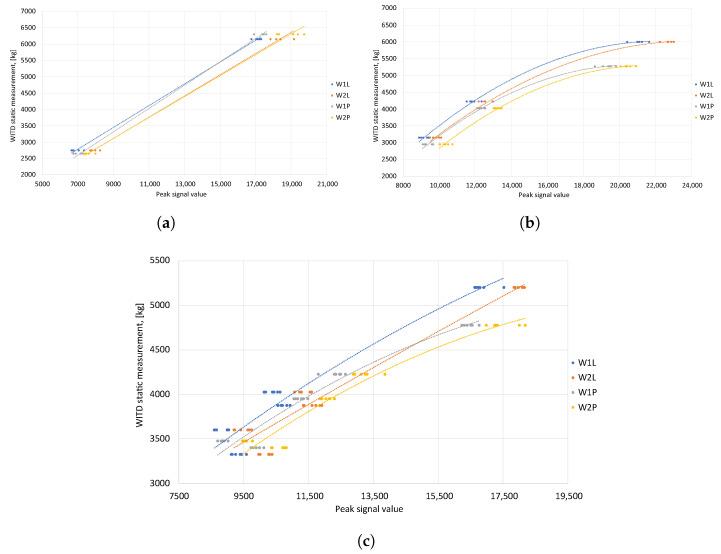
Signal maximum—determination of individual calibration functions for each type of test vehicle. (**a**) 2-axle vehicle—linear approximation. (**b**) 3-axle vehicle—polynomial approximation. (**c**) 5-axle vehicle—polynomial approximation.

**Figure 9 sensors-24-04845-f009:**
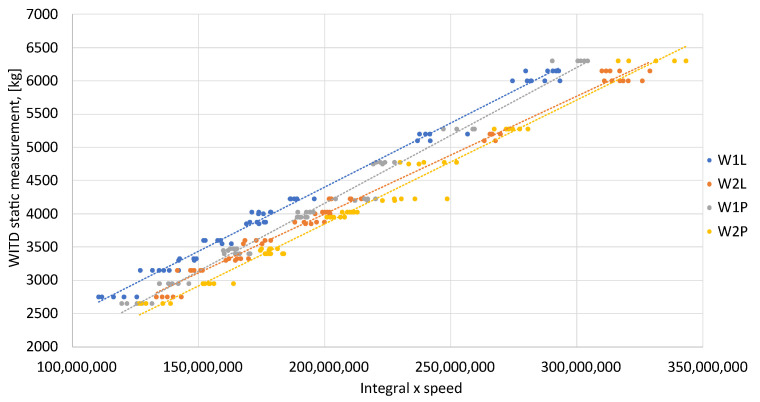
The area under the signal waveform—establishing calibration functions for the complete set of test vehicles. The lines represent linear interpolation.

**Figure 10 sensors-24-04845-f010:**
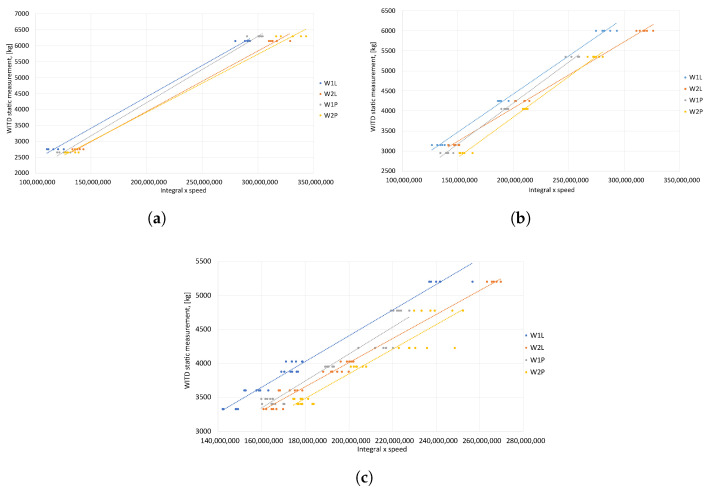
The area under the signal waveform—assignment of specific calibration functions for every test vehicle. (**a**) Two-axle vehicle—linear approximation. (**b**) Three-axle vehicle—linear approximation. (**c**) Five-axle vehicle—linear approximation.

**Figure 11 sensors-24-04845-f011:**
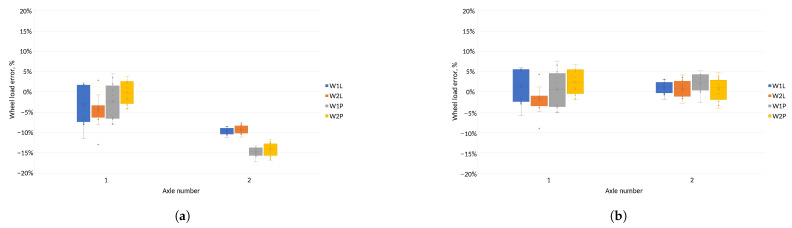
Evaluation of discrepancies in ascertaining the pressures on distinct sensors for shared and individual calibration functions for the technique of determining the aggregate mass based on the peak of the signal waveform. (**a**) Error in estimating the pressure exerted on the sensor—common calibration functions. (**b**) Error in measurement of the pressure exerted on the sensor—calibration functions for the vehicle type. (**c**) Error in estimating the pressure exerted on the sensor—common calibration functions. (**d**) Error in estimating the pressure exerted on the sensor—calibration functions for vehicle type. (**e**) Error in estimating the pressure exerted on the sensor—common calibration functions. (**f**) Error in measurement of the pressure exerted on the sensor—calibration functions for vehicle type.

**Figure 12 sensors-24-04845-f012:**
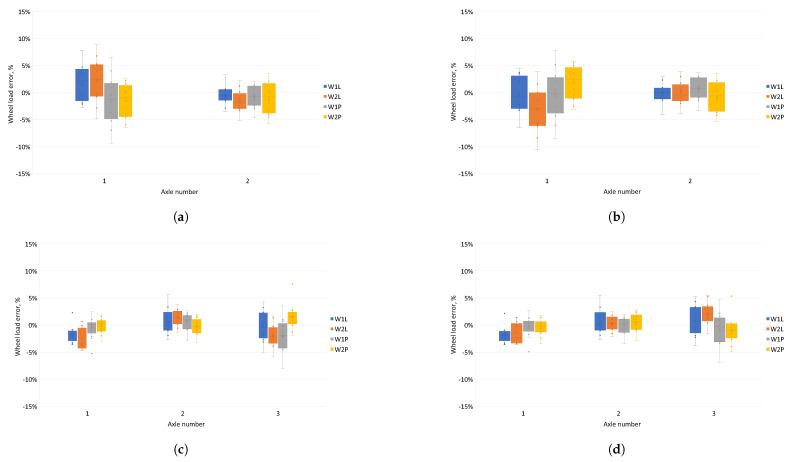
Evaluation of discrepancies in ascertaining the pressures on individual sensors using shared and distinct calibration functions for the technique of calculating the total mass based on the area beneath the signal waveform and the vehicle’s velocity. (**a**) Relative error in determining the pressure on the sensor—common calibration functions. (**b**) Relative error in determining the pressure on the sensor—calibration functions for the vehicle type. (**c**) Relative error in determining the pressure on the sensor—common calibration functions. (**d**) Relative error in determining the pressure on the sensor—calibration functions for the vehicle type. (**e**) Relative error in determining the pressure on the sensor—common calibration functions. (**f**) Relative error in determining the pressure on the sensor—calibration functions for the vehicle type.

**Figure 13 sensors-24-04845-f013:**
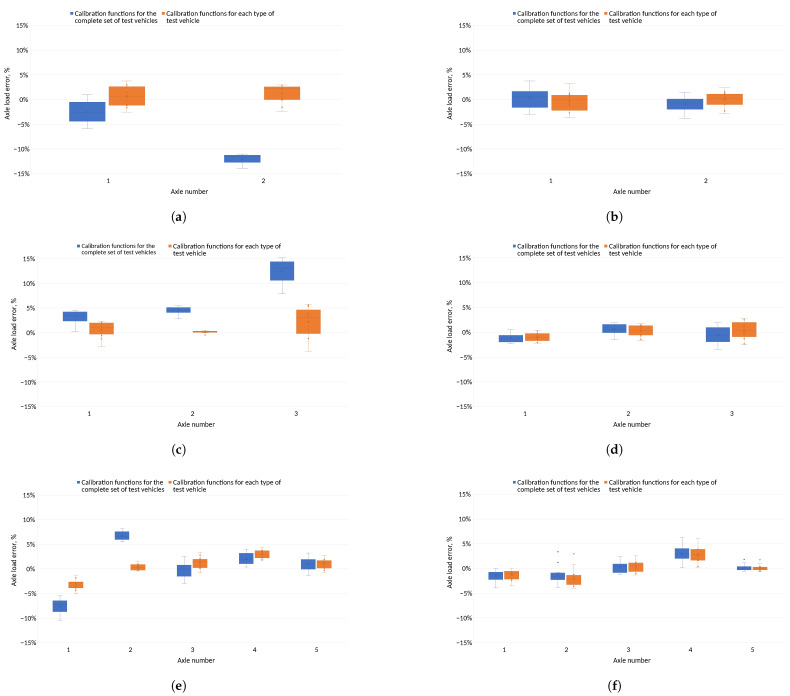
Evaluation of discrepancies in ascertaining axle burdens for shared and distinct calibration functions using two different techniques for determining axle weights. (**a**) Two-axle vehicle, maximum signal—relative error in determining the axle load. (**b**) Two-axle vehicle, area under the signal waveform—relative error in determining the axle load. (**c**) Three-axle vehicle, maximum signal—relative error in determining axle load. (**d**) Three-axle vehicle, area under the signal waveform—relative error in determining the axle load. (**e**) Five-axle vehicle, maximum signal—relative error in determining the axle load. (**f**) Five-axle vehicle, area under the signal waveform—relative error in determining the axle load.

**Figure 14 sensors-24-04845-f014:**
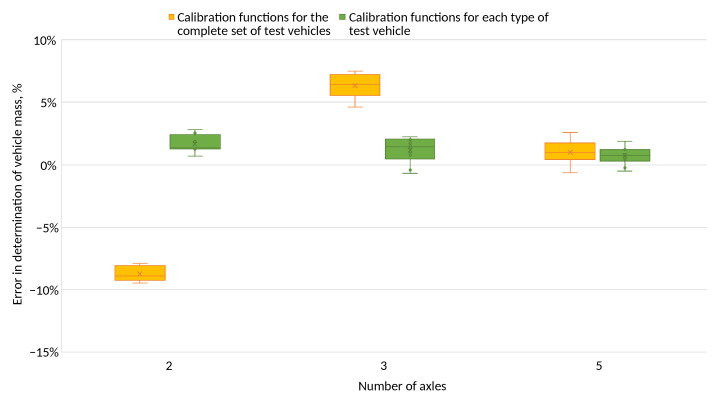
Error in calculating the overall mass—establishing the pressure from the peak value of the sensor’s waveform.

**Figure 15 sensors-24-04845-f015:**
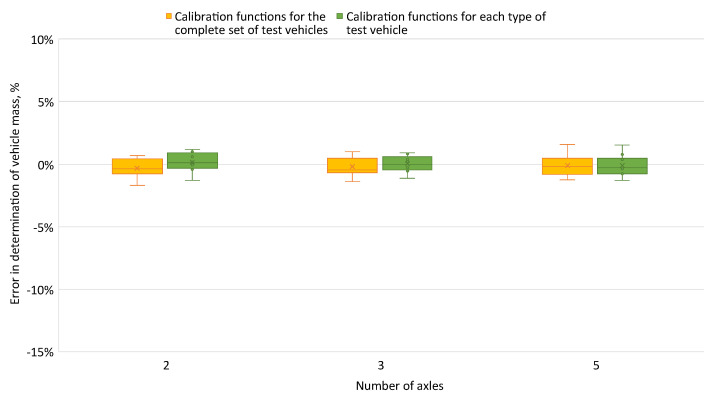
Error in determining the total mass—determining the pressure based on the field under the signal from the sensor.

**Figure 16 sensors-24-04845-f016:**
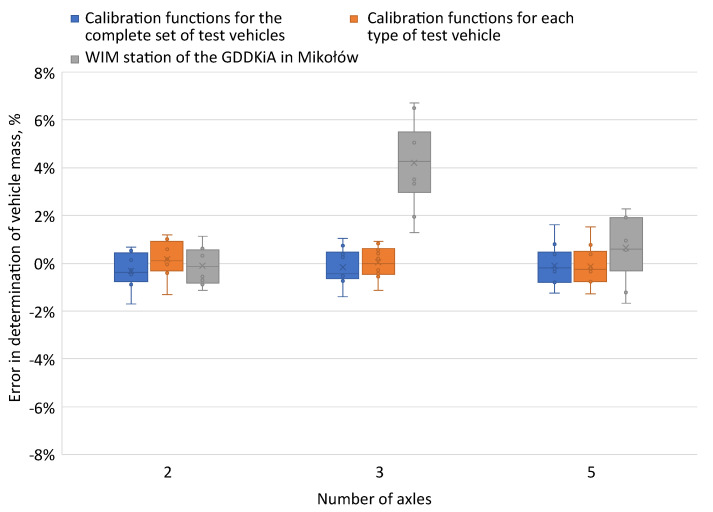
Comparison of errors in determining the total mass of test vehicles for two variants of calibration functions and the preselection weighing station for moving vehicles in Mikołów.

**Table 1 sensors-24-04845-t001:** Measurements of test vehicles using the WITD scale have been referenced.

Vehicle	Axle Number	Mean Value $W_{\mathrm{SL}}$, kg	Mean Value $W_{\mathrm{SP}}$, kg	Axle Press., kg	Total Mass, kg
2-axle	1	2750	2650	5400	17,850
	2	6150	6300	12,450	
3-axle	1	4225	4025	8250	25,625
	2	6000	5275	11,275	
	3	3150	2950	6100	
5-axle	1	3600	3475	7075	39,850
	2	5200	4775	9975	
	3	4025	3400	7425	
	4	3325	4225	7550	
	5	3875	3950	7825	

**Table 2 sensors-24-04845-t002:** Reference measurements of test vehicles with the LP788 scale.

Vehicle	Axle Number	Mean Value $W_{\mathrm{SL}}$, kg	Mean Value $W_{\mathrm{SP}}$, kg	Axle Press., kg	Total Mass, kg
2-axle	1	2862	2779	5641	18,284
	2	6259	6385	12,644	
3-axle	1	4205	4200	8405	27,458
	2	5724	6575	12,298	
	3	3441	3315	6756	
5-axle	1	3762	3550	7275	41,622
	2	4929	5056	9985	
	3	4384	3789	8173	
	4	3544	4518	8061	
	5	4040	4089	8128	

**Table 3 sensors-24-04845-t003:** Calibration functions for each type of test vehicle.

Vehicle Type	Calibration Function	$R^{2}$
Biaxial	$W_{D1L}=0.3326\cdot \max\left(X_{i}\right)+460.2$	1.00
	$W_{D2L}=0.3238\cdot \max\left(X_{i}\right)+198.92$	1.00
	$W_{D1P}=0.3557\cdot \max\left(X_{i}\right)+116.4$	1.00
	$W_{D2P}=0.3187\cdot \max\left(X_{i}\right)+248.7$	0.99
Tri-axial	$W_{D1L}=-0.00001684\cdot {\left(\max\left(X_{i}\right)\right)}^{2}+0.7487\cdot \max\left(X_{i}\right)-2300.$	1.00
	$W_{D2L}=-0.00001414\cdot {\left(\max\left(X_{i}\right)\right)}^{2}+0.6797\cdot \max\left(X_{i}\right)-2139.$	1.00
	$W_{D1P}=-0.00001818\cdot {\left(\max\left(X_{i}\right)\right)}^{2}+0.7558\cdot \max\left(X_{i}\right)-2540.$	1.00
	$W_{D2P}=-0.00001798\cdot {\left(\max\left(X_{i}\right)\right)}^{2}+0.7814\cdot \max\left(X_{i}\right)-3186.$	1.00
Five-axial	$W_{D1L}=-0.00000604\cdot {\left(\max\left(X_{i}\right)\right)}^{2}+0.3717\cdot \max\left(X_{i}\right)+646.4$	0.94
	$W_{D2L}=-0.00000106\cdot {\left(\max\left(X_{i}\right)\right)}^{2}+0.2335\cdot \max\left(X_{i}\right)+1340.$	0.93
	$W_{D1P}=-0.00000924\cdot {\left(\max\left(X_{i}\right)\right)}^{2}+0.4216\cdot \max\left(X_{i}\right)+350.0$	0.94
	$W_{D2P}=-0.00000909\cdot {\left(\max\left(X_{i}\right)\right)}^{2}+0.4263\cdot \max\left(X_{i}\right)+106.0$	0.94

**Table 4 sensors-24-04845-t004:** The area under the signal waveform—specified adjustment processes for every kind of examination vehicle.

Vehicle Type	Calibration Function	$R^{2}$
Biaxial	$W_{D1L}=0.00001973\cdot V_{L}\cdot \int_{t_{1}-t_{\Delta }}^{t_{2}+t_{\Delta }}\left(x\left(t\right)-b\left(t\right)\right)\;dx+449.8$	1.00
	$W_{D2L}=0.00001002\cdot V_{L}\cdot \int_{t_{1}-t_{\Delta }}^{t_{2}+t_{\Delta }}\left(x\left(t\right)-b\left(t\right)\right)\;dx+132.88$	1.00
	$W_{D1P}=0.00002084\cdot V_{P}\cdot \int_{t_{1}-t_{\Delta }}^{t_{2}+t_{\Delta }}\left(x\left(t\right)-b\left(t\right)\right)\;dx+46.32$	1.00
	$W_{D2P}=0.00001827\cdot V_{P}\cdot \int_{t_{1}-t_{\Delta }}^{t_{2}+t_{\Delta }}\left(x\left(t\right)-b\left(t\right)\right)\;dx+258.7$	0.99
Tri-axial	$W_{D1L}=0.00001905\cdot V_{L}\cdot \int_{t_{1}-t_{\Delta }}^{t_{2}+t_{\Delta }}\left(x\left(t\right)-b\left(t\right)\right)\;dx+610.2$	0.99
	$W_{D2L}=0.00001658\cdot V_{L}\cdot \int_{t_{1}-t_{\Delta }}^{t_{2}+t_{\Delta }}\left(x\left(t\right)-b\left(t\right)\right)\;dx+751.3$	0.99
	$W_{D1P}=0.00002052\cdot V_{P}\cdot \int_{t_{1}-t_{\Delta }}^{t_{2}+t_{\Delta }}\left(x\left(t\right)-b\left(t\right)\right)\;dx+92.36$	0.99
	$W_{D2P}=0.00002020\cdot V_{P}\cdot \int_{t_{1}-t_{\Delta }}^{t_{2}+t_{\Delta }}\left(x\left(t\right)-b\left(t\right)\right)\;dx-194.3$	0.99
Five-axial	$W_{D1L}=0.00001889\cdot V_{L}\cdot \int_{t_{1}-t_{\Delta }}^{t_{2}+t_{\Delta }}\left(x\left(t\right)-b\left(t\right)\right)\;dx+625.2$	0.98
	$W_{D2L}=0.00001771\cdot V_{L}\cdot \int_{t_{1}-t_{\Delta }}^{t_{2}+t_{\Delta }}\left(x\left(t\right)-b\left(t\right)\right)\;dx+465.2$	0.99
	$W_{D1P}=0.00001982\cdot V_{P}\cdot \int_{t_{1}-t_{\Delta }}^{t_{2}+t_{\Delta }}\left(x\left(t\right)-b\left(t\right)\right)\;dx+171.0$	0.92
	$W_{D2P}=0.00001809\cdot V_{P}\cdot \int_{t_{1}-t_{\Delta }}^{t_{2}+t_{\Delta }}\left(x\left(t\right)-b\left(t\right)\right)\;dx+225.9$	0.89

**Table 5 sensors-24-04845-t005:** Comparison of calibration and verification runs using the peak signal—determination of common calibration functions based on the median of the total mass and error in relation to reference measurements.

	Calibration Runs		Check-Ups	
Vehicle Type	Median $W_{D}$, kg	Error δ	Median $W_{D}$, kg	Error δ
Biaxial	16,170	−9%	16,300	−9%
Tri-axial	26,950	5%	27,410	7%
Five-axial	39,760	0%	40,360	1%

**Table 6 sensors-24-04845-t006:** The median of the overall weight and discrepancy compared to benchmark measurements for specific calibration functions was ascertained using the peak of the signal—evaluation of calibration and validation processes.

	Calibration Runs		Check-Ups	
Vehicle Type	Median $W_{D}$, kg	Error δ	Median $W_{D}$, kg	Error δ
Biaxial	17,951	1%	18,172	2%
Tri-axial	25,604	0%	26,075	2%
Five-axial	39,762	0%	40,114	1%

**Table 7 sensors-24-04845-t007:** Median and error of the total mass relative to reference measurements, ascertained using the area under the signal waveform for typical calibration functions—evaluation of calibration and validation processes.

	Calibration Runs		Check-Ups	
Vehicle Type	Median $W_{D}$, kg	Error δ	Median $W_{D}$, kg	Error δ
Biaxial	17,872	0%	17,766	−2%
Triaxial	25,489	−1%	25,508	0%
Five-axial	39,845	0%	39,742	0%

**Table 8 sensors-24-04845-t008:** Median of the overall weight and discrepancy compared to benchmark measurements for each calibration function for a specific vehicle type, ascertained using the area beneath the signal waveform—evaluation of calibration and validation processes.

	Calibration Runs		Check-Ups	
Vehicle Type	Median $W_{D}$, kg	Error δ	Median $W_{D}$, kg	Error δ
Biaxial	17,956	1%	17,843	0%
Triaxial	25,719	0%	25,698	0%
Five-axial	39,823	0%	39,742	0%

**Table 9 sensors-24-04845-t009:** Maximum error in determining the axle load calculated on the basis of the peak value of the signal at different speeds.

Vehicle Type	Common Calibration Functions			Specific Calibration Functions		
	40 km/h	50 km/h	60 km/h	40 km/h	50 km/h	60 km/h
Biaxial	11%	13%	11%	3%	3%	3%
Tri-axial	14%	15%	15%	4%	6%	6%
Five-axial	8%	8%	8%	4%	4%	4%

**Table 10 sensors-24-04845-t010:** Maximum error in determining the axle load calculated using the field under the signal waveform for various speeds.

Vehicle Type	Common Calibration Functions			Specific Calibration Functions		
	40 km/h	50 km/h	60 km/h	40 km/h	50 km/h	60 km/h
Biaxial	2%	3%	2%	1%	2%	3%
Tri-axial	3%	2%	2%	3%	2%	2%
Five-axial	2%	4%	4%	3%	4%	4%

**Table 11 sensors-24-04845-t011:** Maximum signal—errors in determining the loads on the axle and axle group.

Vehicle Type	Common Calibration Functions		Calibration Function Depends on the Vehicle Type	
	Maximum Error per Axis	Maximum Error per Axis of the Axis Group	Maximum Error per Axis	Maximum Error per Axis of the Axis Group
Biaxial	14%	-	4%	-
Triaxial	0%	15%	6%	6%
Five-axial	10%	4%	5%	4%

**Table 12 sensors-24-04845-t012:** Area under the signal waveform—errors in determining axle loads and axle groups.

Vehicle Type	Common Calibration Functions		Calibration Function Depends on the Vehicle Type	
	Maximum Error per Axis	Maximum Error per Axis of the Axis Group	Maximum Error per Axis	Maximum Error per Axis of the Axis Group
Biaxial	4%	-	4%	-
Triaxial	2%	3%	2%	3%
Five-axial	4%	6%	4%	6%

## Data Availability

Data can be obtained from corresponding author via e-mail.

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
