# Peer review of "Strain Gauge Calibration for High Speed Weight-in-Motion Station"

_sensors, 2024, doi:10.3390/s24154845_

Round 1

Reviewer 1 Report

Comments and Suggestions for Authors

The manuscript introduces two calibration methods of HSWIM systems for determining the wheel load, which are based on the maximum of the signal from strain gauge sensors and for the method using the field under the signal and vehicle speed, respectively. The test results shows that the two proposed methods could improve the accuracy of weighing in motion, which did not require any interference with the measurement station. Because the proposed  calibration methods benefit the development of HSWIM systems compared with the published papers ,the conclusions are consistent with evidence ,and the manuscript has the appropriate references, I recommend this manuscript to be published in Sensor Journal after the following revision.

1. On table I of Page 6 and table 2 of page 7, “Axle press., kG” should be “Axle press, kg”

2. On fig.5 of page 8, the author should show the meaning of data in red and blue color as figure caption.

3. On Eq.(5) of page 11, the authors should indicate the meaning of t1,t2, t_delta, b(t) below the equation in detail. The use of the similar symbol b(t) and b in Eq.(5) will confuse the reader.

4.On lines 404-406 of page 15 , “In the case of determining the wheel loads and total mass based on the area under the signal waveform, errors relative to the reference measurement remain at a similar level for both standard and individual calibration functions; Table 7, Table 8.” should be changed into “In the case of determining the wheel loads and total mass based on the area under the signal waveform, errors relative to the reference measurement remain at a similar level for both standard and individual calibration functions shown in Table 7 and Table 8.” 

Author Response

Comment 1. On table I of Page 6 and table 2 of page 7, “Axle press., kG” should be “Axle press, kg” 

Dear reviewer, the text in table 2 has been corrected. 

Comment 2. On fig.5 of page 8, the author should show the meaning of data in red and blue color as figure caption. 

We completed the drawing caption and placed a legend on the chart. 

Comment 3. On Eq.(5) of page 11, the authors should indicate the meaning of t1,t2, t_delta, b(t) below the equation in detail. The use of the similar symbol b(t) and b in Eq.(5) will confuse the reader.

The description of the equation has been corrected; parameter b has been replaced with parameter C2. The description of the equation was supplemented with the meaning of the parameters t1, t2, t_delta, b(t), which appear in the equation. 

Comment 4.On lines 404-406 of page 15 , “In the case of determining the wheel loads and total mass based on the area under the signal waveform, errors relative to the reference measurement remain at a similar level for both standard and individual calibration functions; Table 7, Table 8.” should be changed into “In the case of determining the wheel loads and total mass based on the area under the signal waveform, errors relative to the reference measurement remain at a similar level for both standard and individual calibration functions shown in Table 7 and Table 8.” 

This text has been corrected. 

Reviewer 2 Report

Comments and Suggestions for Authors

High-speed WIM technology is very promising in the field of road supervision. In this paper, the influence of two data processing methods on the weighing error of the data of the strain gauge WIM sensor is studied. The article needs to be further refined and improved:

(1)         The abbreviation that appears for the first time in the article needs to be explained, such as GVW WITED, etc.

(2)         It is recommended to supplement the principle and structural form, main parameters, and installation and fixing methods of WIM sensors

(3)         This paper lacks an analysis of the influence of vehicle speed on WIM accuracy. High-speed WIM with a defined speed range of 30 km/h to 130 km/h, The impact of vehicle speed on measurement accuracy is significant.

(4)         The article has multiple fitting equations and curves, and R2 needs to be indicated;

(5)         As shown in Figure 4, there are four established sensors, and each sensor tests one side wheel load. So, The verification needs to be the wheel load. It is recommended to supplement the analysis of how the mass error of a single wheel and the error of the vehicle's total mass is transmitted.

(6)         The article lacks an analysis of the causes of measurement errors.

Author Response

Comment ( 1)The abbreviation that appears for the first time in the article needs to be explained, such as GVW WITED, etc.

Dear Reviewer, we have supplemented our text with an explanation of all abbreviations and acronyms. 

Comment (2)It is recommended to supplement the principle and structural form, main parameters, and installation and fixing methods of WIM sensors  -

The description of the system operation in point 2.2 has been extended and the description of its elements has been corrected. Now it reads as follows: 

The block diagram of the weighing in-motion system in MikoÅ‚ów is shown in figure 1. Strain gauge sensors, piezoelectric sensors and inductive loops installed in the road surface are connected to the iWIM data logger. 

Figure 1. Block diagram of WIM system in MikoÅ‚ów [19]. 

The data collected by the components of the WIM station undergo processing in the analog and digital pathways of the iWIM data logger. In the analogue phase, the signals from the sensors are sampled at a frequency of several tens of kilohertz and undergo initial filtering. Subsequently, in the digital phase, an FPGA module processes the data, which comprises functional blocks such as sensor input, decimation and compensation filters, circular buffer, and estimator. The sensor input module is responsible for receiving data from the measurement amplifiers. Signals received from sensors undergo filtration, the sampling frequency is reduced in the decimation filter, and any frequency discrepancies are corrected in the compensation filter. The data obtained are stored in a circular buffer and undergo preliminary analysis in the estimator module. Using data from induction loops and pressure sensors, the estimator predicts the vehicle speed and the entry and exit times at the measurement station. In addition, details such as the maximum signal values and vehicle ID are stored in the circular buffer. Specialized software identifies the signal peaks corresponding to the vehicle wheels passing and calculates the integral of the recorded signal. The exact vehicle speed and axle distances are determined by analyzing the location of the highest signal values. Subsequently, the iWIM data logger calculates the total weight of the vehicle on the basis of the applied algorithm. The ALPR camera, the overview camera and the iWIM data logger transmit data to the WIM roadside unit on which the dedicated WIM system software is implemented. The software of the roadside unit includes modules that operate the specific devices, a time server and a module of processing the collected data. The host software generates a data file that contains all recorded vehicle information. The precision of the collected data is then evaluated using the method outlined in [18,19]. A data record of a vehicle is encrypted and sent to another software module on the WIM server using the VPN. The data from the WIM server is then passed on to a graphical user interface (GUI) in an online application, where the collected data of each vehicle are visualised [18,19]. 

Comment (3)This paper lacks an analysis of the influence of vehicle speed on WIM accuracy. High-speed WIM with a defined speed range of 30 km/h to 130 km/h, The impact of vehicle speed on measurement accuracy is significant. 

Dear reviewer, you are right when you write that the measurements should be within the speeds permitted on public roads. However, in the case of our research, it was impossible to examine the impact of speed changes over such a wide range. The WIM station that we built is located on a road with a maximum speed of 50 km/h during regular vehicle traffic. The traffic conditions in which the measurements were carried out allowed the test vehicles to travel at a maximum speed of 60 km/h. In addition, the measurements were carried out as part of periodic calibration, which determines the speeds at which test vehicles travel. Drivers of test vehicles and WITD employees were limited by their allowed working time, and the number of test runs had to be adapted to their availability and traffic conditions. 

Comment (4)The article has multiple fitting equations and curves, and R2 needs to be indicated;

The values ​​of the R2 parameter are given for each equation. 

Comment(5)As shown in Figure 4, there are four established sensors, and each sensor tests one side wheel load. So, The verification needs to be the wheel load. It is recommended to supplement the analysis of how the mass error of a single wheel and the error of the vehicle's total mass is transmitted.

Additional analysis is presented in the Conclusions section. Now it reads as follows: 

Weigh in motion systems, and in particular HSWIM systems, are an element that allows the identification of overloaded vehicles from the traffic flow. The development of on-the-go weighing systems towards automatic mandates requires the appropriate selection of an algorithm for assessing the reliability of measurements and increasing the measurement accuracy to the required level. 

An extremely important element is the appropriate determination of the reference level, because it is on its basis that calibration functions are determined. The accuracy of the static measurement is 2%. An additional factor affecting the accuracy of this measurement is the time that has elapsed since the portable wheel scales were calibrated. The degree of degradation of the administrative weighing station can also distort the measurement, through the difference in levels between the weighed axis and the other axes. During the tests, a difference in 2 cm levels caused significant differences between the static measurements. 

Figures 11,12 show the errors of determining the wheel pressures on the sensor. It can be seen in these figures that there is a scatter of errors, i.e. some of the errors are positive and some negative. Determining the axle pressures as averages of the measurements on the two lines results in the errors in the determination of the axle pressures (figure 13) and the total vehicle weight (figure 14, figure 15) being significantly smaller than the error of sensor pressures. Implementing calibration functions for the vehicle type stabilises the wheel pressure determination errors on the sensor. Increasing the measurement accuracy on the sensors will enable the detection of vehicles that are unevenly loaded. 

The determination of wheel pressures using the peak value of the signal does not take speed into account in the calculation and is used for systems where a high degree of accuracy is not required. For more accurate systems, e.g. pre-selection, the second method is used, where the calculation takes vehicle speed into account. 

The road surface in which the load sensors are installed is a part of the measuring system is constantly degraded by vehicle traffic. An increase in the pressure error value of a sensor or a pair of sensors, e.g. the left sensors, can be caused by the fluctuations of a vehicle passing through ruts in the roadway. 

Comment(6)The article lacks an analysis of the causes of measurement errors. -

Additional explanations are provided in the Conclusions section. 

Reviewer 3 Report

Comments and Suggestions for Authors

Greetings to the authors of the article, it is a nice and useful research topic. The presented method of calibrating the sensor system seems promising to me, because it can also be used for other techniques for measuring strain, e.g. optical fiber Bragg grating sensors.

I have some suggestions/questions to improve the article:

- I miss a description of the function of the used strain sensing WIM system

- the axes of the graph in Figure 5 lack a description. I assume that it is the strain's dependence on the measurement time? Then I think the value of almost 20,000 microstrains is very high, especially from the point of view of the lifetime of the strain gauge itself.

And a few notes on formatting:

- from a typographic point of view, there cannot be two headings in a row without any text between them

- there is an unexplained abbreviation in the abstract (HSWIM)

As for English, I don't feel qualified to judge.

Author Response

Comment 1. I miss a description of the function of the used strain sensing WIM system

The description of the system operation in point 2.2 has been extended and the description of its elements has been corrected. Now it reads as follows:  

The block diagram of the weighing in-motion system in MikoÅ‚ów is shown in Figure 1. Strain gauge sensors, piezoelectric sensors and inductive loops installed in the road surface are connected to the iWIM data logger.  

Figure 1. Block diagram of WIM system in MikoÅ‚ów [19]. 

The data collected by the components of the WIM station undergo processing in the analog and digital pathways of the iWIM data logger. In the analogue phase, the signals from the sensors are sampled at a frequency of several tens of kilohertz and undergo initial filtering. Subsequently, in the digital phase, an FPGA module processes the data, which comprises functional blocks such as sensor input, decimation and compensation filters, circular buffer, and estimator. The sensor input module is responsible for receiving data from the measurement amplifiers. Signals received from sensors undergo filtration, the sampling frequency is reduced in the decimation filter, and any frequency discrepancies are corrected in the compensation filter. The data obtained are stored in a circular buffer and undergo preliminary analysis in the estimator module. Using data from induction loops and pressure sensors, the estimator predicts the vehicle speed and the entry and exit times at the measurement station. In addition, details such as the maximum signal values and vehicle ID are stored in the circular buffer. Specialized software identifies the signal peaks corresponding to the vehicle wheels passing and calculates the integral of the recorded signal. The exact vehicle speed and axle distances are determined by analyzing the location of the highest signal values. Subsequently, the iWIM data logger calculates the total weight of the vehicle on the basis of the applied algorithm.  The ALPR camera, the overview camera and the iWIM data logger transmit data to the WIM roadside unit on which the dedicated WIM system software is implemented. The software of the roadside unit includes modules that operate the specific devices, a time server and a module of processing the collected data. The host software generates a data file that contains all recorded vehicle information. The precision of the collected data is then evaluated using the method outlined in [18,19]. A data record of a vehicle is encrypted and sent to another software module on the WIM server using the VPN. The data from the WIM server is then passed on to a graphical user interface (GUI) in an online application, where the collected data of each vehicle are visualised [18,19]. 

Comment 2. the axes of the graph in Figure 5 lack a description. I assume that it is the strain's dependence on the measurement time? Then I think the value of almost 20,000 microstrains is very high, especially from the point of view of the lifetime of the strain gauge itself. .

Dear Reviewer, the figure has been supplemented with a description of the axis, a legend has been placed on the graph, and the description of the graph has been extended. 

And a few notes on formatting: 

Comment 3. from a typographic point of view, there cannot be two headings in a row without any text between them -

Dear Reviewer, the layout of the headings has been modified. 

- there is an unexplained abbreviation in the abstract (HSWIM) - Dear Reviewer, the Abstract now explains the abbreviation HSWIM. Other previously unexplained abbreviations have also been expanded. 

Round 2

Reviewer 2 Report

Comments and Suggestions for Authors

This paper investigated the impact of two different data processing methods on the accuracy of the HSWIM system and analyze the impact of different vehicle models on the results. The experimental design is reasonable, the data analysis is reliable and the conclusions and suggestions have certain reference value. It is suggested that the author should make revisions and additions according to the reviewers' comments to further improve the quality of the article.

1.      It is recommended to supplement the analysis of the influence of vehicle speed on the accuracy of the WIM system, test the accuracy of the WIM system at different speeds, and analyze the mechanism of its influence.

2.      It is recommended to supplement the description of sensor type, structure, parameters, etc. You can refer to the information provided by the sensor manufacturer to describe the type, structure, and parameters of the sensor in detail, so that readers can better understand the experimental process.

3.      It is recommended to put forward more specific conclusions and suggestions for different application scenarios, such as selecting the appropriate calibration function according to the vehicle model, improving the robustness of the WIM system, etc.

Author Response

Comment 1: It is recommended to supplement the analysis of the influence of vehicle speed on the accuracy of the WIM system, test the accuracy of the WIM system at different speeds, and analyze the mechanism of its influence. 

Dear esteemed reviewer, 

In section 1.2.3, the authors have referenced sources [14,15] that delve into the impact of the vehicle's braking and acceleration as it traverses the measurement station. Furthermore, these sources shed light on the significant influence of the road surface condition on the outcomes of measurements conducted using a strain gauge scale.  Please take note of the following: In a new section 4.3, you will find a new tables 9 and 10, which illustrate the maximum errors in determining the axle load for various speeds. Additionally, the second method of determining pressure incorporates speed into the pressure equations. 

Comment 2: t is recommended to supplement the description of sensor type, structure, parameters, etc. You can refer to the information provided by the sensor manufacturer to describe the type, structure, and parameters of the sensor in detail, so that readers can better understand the experimental process.

Dear reviewer, the parameters characterizing the strain gauge sensor have been supplemented in points 2.1 and 3.2 based on an article published by the sensor manufacturer. In point 2.1 we have following text added: 

“The structure of the sensor makes it adapted to measure only the vertical force acting on the sensor. Strain gauge sensors are not sensitive to lateral forces occurring through the action of the road surface on the sensor. Furthermore, the sensors used are equipped with an internal temperature compensation circuit so that they are not sensitive to changes in pavement temperature and the measurements are stable over time \cite{ref-journal15}. In addition, they are characterised by a low linearity error of <0.1± \%\ FSO.” 

Comment 3: 

  It is recommended to put forward more specific conclusions and suggestions for different application scenarios, such as selecting the appropriate calibration function according to the vehicle model, improving the robustness of the WIM system, etc. 

Dear reviewer, the conclusions have been supplemented with the content requested by you. We read there: 

"When comparing the maximum values of axle load errors at the tested speeds (see table \9}, table 10), it's evident that the differences for the tested speeds are approximately 1%. Furthermore, the second method of determining pressures, involving the utilization of the area under the signal waveform, considers the vehicle speed in the equations, leading to a more comprehensive analysis. It's crucial to acknowledge that under regular road traffic conditions, conducting measurements is subject to limitations imposed by other road users.” 

and further:

"The test vehicles used have varying loads on individual axles. By employing calibration functions, we can precisely adjust the pressure equation to suit each vehicle type, resulting in accurate pressure determination and total weight measurement across all vehicles. Additionally, the number of wheels impacts the number of samples used to determine standard calibration functions. The most favourable outcomes are achieved for a five-axle vehicle. The utilization of type-specific calibration functions ensures consistent accuracy in determining axle pressures for all vehicles. Due to the use of calibration functions for a given vehicle type, errors in determining the total mass remained at a similar level for each type of test vehicle. The improvement achieved in the accuracy of determining the axle loads and total weight allows the tested station to be classified as an automatic system. An accuracy improvement was achieved for both the method using the maximum signal from the sensor and the method based on the field under the signal and the vehicle speed. The proposed solution does not require any interference with the measurement station."
